

# Addressing challenges in uncertainty quantification. The case of geohazard assessments

Ibsen Chivata Cardenas[1], Terje Aven[1], Roger Flage[1]

1Department of Safety, Economics and Planning, University of Stavanger, Stavanger, 4021, Norway

*Correspondence to:* Ibsen Chivata Cardenas (ibsen.chivatacardenas@uis.no)

**Abstract.** By describing critical tasks in quantifying uncertainty using geohazard models, we analyse some of the challenges involved. Under the often-seen condition of very limited data and despite the availability of recently developed sophistications to parameterise models, a major challenge that remains is the constraining of the many model parameters involved. However, challenges also lie in the credibility of predictions required in the

assessments, the uncertainty of input quantities, and the conditional nature of the quantification on the choices and assumptions made by analysts. Addressing these challenges calls for more insightful approaches that are yet to be developed; however, clarifications and reinterpretations of some fundamental concepts together with practical simplifications may be required first, and these are discussed in this paper. The research aims at strengthening both the foundation of geohazard risk assessments and its practice.

## 15   1 Introduction

Uncertainty quantification, UQ, helps determine the uncertainty of a system's responses when some quantities and events in such system are unknown. Using models, system's responses can be calculated analytically, numerically, or by random sampling (Monte Carlo method, rejection sampling, Monte Carlo sampling using Markov chains, importance sampling, subset simulation) (Metropolis and Ulam, 1949; Brown, 1953; Ulam, 1961; Hastings, 1970).

Given the high-dimensional and spatial nature of hazard events and associated quantities, sampling methods are frequently used because they result in a less expensive and more tractable uncertainty quantification in comparison with analytical and numerical methods. In the sampling procedure, specified distributions of the input quantities and parameters are sampled and respective outputs of the model are recorded, then the process is repeated as many times as may be required to achieve the desired accuracy (Vanmarcke, 1984). Eventually, the distribution of the

outputs can be used to calculate probability-based metrics, such as expectations or probabilities of critical events. Model-based uncertainty quantification using sampling is now more often used in geohazard assessments, e.g., Uzielli and Lacasse (2007), Wellmann and Regenauer-Lieb (2012), Rodríguez-Ochoa et al. (2015), Pakyuz-Charrier et al. (2018), Huang et al. (2021), Luo et al. (2021), Sun et al. (2021a).

In this paper, we consider recent advances in UQ and analyse some remaining challenges. For instance, we note

that, despite the availability of sophistications to parameterise models used for UQ, a major problem persists, namely the constraining of the many parameters involved. In practice, based solely on historical data, only some parameters can be constrained (e.g., Albert, Callies, and von Toussaint, 2022). Another challenge is that model outputs are not only conditional on the choice of model parameters, but also on input quantities, including initial and boundary conditions. For example, a geological system model could be specified to include some structures

in the ground and geological boundary conditions (Juang et al., 2019). Such systems are usually time dependent and spatial in nature involving, e.g., possibly changing conditions (e.g., Chow, Li, and Koh, 2019). Incorporating




uncertainties related to such conditions complicates the modelling and demands further acquisition of data. Next, models could be accurate at reproducing data from past events but may be inadequate for unobserved outputs or predictions, as might be the case when predicting, e.g., extreme velocities in marine turbidity currents, which are

driven by emerging and little understood soil and fluid interactions (Vanneste et al., 2019). Overlooking these challenges in a geohazard assessment implies that the quantification will only reflect some aspects of the uncertainty involved. These challenges are, unfortunately, neither exhaustively nor clearly discussed in the geohazard literature. Options and clarifications addressing these challenges are underreported in the field, yet analysing these challenges can be useful in treating uncertainties consistently and providing meaningful results in

an assessment. This paper's aim is thus to bridge the gap in the literature by providing an analysis and clarifications enabling a useful quantification of uncertainty.

It should be emphasised that, in this paper, we consider uncertainty quantification in terms of probabilities. Other approaches to measure or represent uncertainty that have been studied by, for example, Zadeh (1968), Shafer (1976), Ferson and Ginzburg (1996), Helton and Oberkampf (2004), Dubois (2006), Aven (2010), Flage et al.,

(2013), Shortridge, Aven, and Guikema (2017), Flage, Aven, and Berner (2018), and Gray et al. (2022a,b), will not be discussed here. The discussion about the complications in UQ related to computational issues generated by numerical approximations including, for instance, sampling procedures are also beyond the scope of the current work.

The remainder of the paper is as follows. In Section 2, based on recent advances, we describe how uncertainty

quantification using geohazard models can be conducted. Next, some remaining challenges in UQ are identified and illustrated. Options to address the challenges in UQ are discussed in Section 3. A simplified example, further illustrating the discussion, is found in Section 4, while the final section provides some conclusions from this study.

## 2 Quantifying uncertainty using geohazard models

In this section, we make explicit critical steps in uncertainty quantification, UQ. We describe a general approach

to UQ that considers uncertainty as the analysts' incomplete knowledge about quantities or events. The UQ approach described is restricted to probabilistic analysis. Emphasis is made on the choices and assumptions usually made by analysts.

A geohazard model can be described as follows. We consider a system (e.g., debris flow) with a set of input quantities $X$ (e.g., sediment concentration, entrainment rate) whose relationships to the output quantities $Y$ (e.g.,

runout volume, velocity, or height of flow) can be expressed by a set of models $\mathcal{M}$. Analysts *identify or specify $X$*, $Y$, and $\mathcal{M}$. A vector $\Theta_{\eta}$ (including, e.g., friction, viscosity, turbulence coefficients) parameterises a model $\eta$ in $\mathcal{M}$. The parameters $\Theta_{\eta}$ determine specific functions among a family of potential functions modelling the system. Accordingly, a model $\eta$ can be described as a multi-output function with, e.g., $Y$ = {runout volume, velocity, height of flow}, and we can write Eq. (1) based on Lu and Lermusiaux (2021):

$$\eta: X_{s,t} \times \Theta_{\eta} \rightarrow Y_{s,t} \tag{1}$$

$$\eta \equiv (E_{\eta}, SG_{\eta}, BC_{\eta}, IC_{\eta}) \tag{2}$$

where $y$ as realisations of $Y$ are the model responses when elements in $X$ take the values $x$ at a spatial location $s \in S$ and a specific time $t \in T$, and parameters $\theta_{\eta} \in \Theta_{\eta}$ are used. In Eq. (1), $X \subset \mathbb{R}^{d_X}$ is the set of input quantities, $T \subset \mathbb{R}^{d_T}$ is the time domain, $S \subset \mathbb{R}^{d_S}$ is the spatial domain, $\Theta_{\eta} \subset \mathbb{R}^{d_{\theta_{\eta}}}$ corresponds to a parameter vector, and





$Y \subset \mathbb{R}^{d_Y}$ is the set of output quantities, with $d = 0, 1, 2,$ or $3$. The system is fully described if $\mathfrak{m}$ is *specified* in terms of a set of equations $\boldsymbol{E_\mathfrak{m}}$ (e.g., conservation equations), the spatial domain geometry $\boldsymbol{SG_\mathfrak{m}}$ (e.g., extension, soil structure), the boundary conditions $\boldsymbol{BC_\mathfrak{m}}$ (e.g., downstream flow), and the initial conditions $\boldsymbol{IC_\mathfrak{m}}$ (e.g., flow at $t = t_0$), see Eq. (2).

In the sampling approach to uncertainty quantification, *specified* probability distributions reflecting analysts'
uncertainty about input quantities are sampled many times, and the distribution of the produced outputs can be calculated. The output probability distribution for a model $\mathfrak{m}$ can be denoted as $f(y|x,\theta_\mathfrak{m},\mathfrak{m})$ for realisations $y, x, \theta_\mathfrak{m}$, $\mathfrak{m}$ of $\boldsymbol{Y}, \boldsymbol{X}, \boldsymbol{\Theta_\mathfrak{m}}$, and $\boldsymbol{M}$, respectively.

Betz (2017) has suggested that the parameter set is fully described by a parameter vector $\boldsymbol{\Theta}$ in Eq. (3):

$$\boldsymbol{\Theta} = \{\boldsymbol{\Theta_\mathfrak{m}}, \boldsymbol{\Theta_X}, \boldsymbol{\Theta_\varepsilon}, \boldsymbol{\Theta_o}\} \tag{3}$$

in which, $\boldsymbol{\Theta_\mathfrak{m}}$ relates to parameters of the model $\mathfrak{m}$, $\boldsymbol{\Theta_X}$ are parameters linked to the input $\boldsymbol{X}$, $\boldsymbol{\Theta_\varepsilon}$ is the vector of the output-prediction error $\varepsilon$, and $\boldsymbol{\Theta_o}$ is the vector associated with observation/measurement errors, when historical records are used. More explicitly, to compute an overall joint probability distribution, we may have the following distributions:

- $f(y|x,\theta_\mathfrak{m},\mathfrak{m})$ is the distribution of $\boldsymbol{Y}$ when $\boldsymbol{X}$ takes the values $x$, and parameters $\theta_\mathfrak{m} \in \boldsymbol{\Theta_\mathfrak{m}}$ and a model $\mathfrak{m} \in$
$\boldsymbol{M}$ are used to compute $y$;

- $f(x|\theta_X,\mathfrak{m})$ is the conditional distribution of $\boldsymbol{X}$ given the parameters $\theta_X \in \boldsymbol{\Theta_X}$ and the model $\mathfrak{m}$. Note that each $\mathfrak{m}$ defines which elements in $\boldsymbol{X}$ are to be considered in the analysis;

- $f(x|\hat{x},\theta_o)$ is a distribution of $\boldsymbol{X}$ given the observed quantities $\boldsymbol{\hat{X}} = \hat{x}$ and the observations/measurement error parameters $\theta_o \in \boldsymbol{\Theta_o}$;

- additionally, one can consider $f(y*|y,\theta_\varepsilon,\mathfrak{m})$, which is a distribution of $\boldsymbol{Y*}$, the future system's response, conditioned on the model output $y$ and the output-prediction error vector $\theta_\varepsilon \in \boldsymbol{\Theta_\varepsilon}$. The output-prediction error $\varepsilon$ is the mismatch between the model predictions and non-observed system's responses $y*$. $\varepsilon$ is used to correct the imperfect model output $y$ (Betz, 2017; Juang et al., 2019).

If, for example, the parameters $\boldsymbol{\Theta_\mathfrak{m}}$ are poorly known, a prior distribution $\pi(\theta_\mathfrak{m}|\mathfrak{m})$ weighing each parameter value
$\theta_\mathfrak{m}$ for a model $\mathfrak{m}$ is usually *specified*. A prior is a subjective probability distribution quantified by expert judgement that represents uncertainty about the parameters prior to considering information in data (Raices-Cruz, Troffaes, and Sahlin, 2022). When some data is available, e.g., in the form of measurements $d = \{\boldsymbol{\hat{Y}} = \hat{y}, \boldsymbol{\hat{X}} = \hat{x}\}$, as a part of different sources of data, i.e., $d \in \boldsymbol{D}$, such parameter values $\theta_\mathfrak{m}$ or their distributions $\pi(\theta_\mathfrak{m}|\mathfrak{m})$ can be constrained by back-analysis methods. Back analysis methods include matching experimental measurements $\hat{y}$ and calculated
model outputs $y$ using different assumed values $\theta'_\mathfrak{m}$, i.e., more formally, values for $\theta_\mathfrak{m}$ can be calculated as follows (based on Liu et al., 2022):

$$\theta_\mathfrak{m} = argmin[\hat{y} - y(\hat{x}, \theta'_\mathfrak{m})] \tag{4}$$

The revision or updating of the prior $\pi(\theta_\mathfrak{m}|\mathfrak{m})$ with data $d$ to obtain a posterior distribution described as $\pi(\theta_\mathfrak{m}|d,\mathfrak{m})$ is also an option in back analysis. The updating can be calculated as follows (based on Juang et al., 2019; Liu et
al., 2022):

$$\pi(\theta_{\boldsymbol{m}}|d, \mathfrak{m}) = \frac{\mathcal{L}(\theta_{\boldsymbol{m}}|d)\pi(\theta_{\boldsymbol{m}}|\mathfrak{m})}{\int \mathcal{L}(\theta_{\boldsymbol{m}}|d)\pi(\theta_{\boldsymbol{m}}|\mathfrak{m})d\theta_{\boldsymbol{m}}} \tag{5}$$

where $\mathcal{L}(\theta_m|d) = f(d|\theta_m)$ is a likelihood function, a distribution which weighs $d$ given $\theta_\mathfrak{m}$.




Similarly, we can constrain any of the distributions above, e.g., $f(y|x,\theta_m,m)$, or $f(x|\theta_X,m)$ to obtain $f(y|x,\theta_m,d,m)$ and $f(x|\theta_X,d,m)$, respectively.

For a geohazard problem, it is possible to *specify* several competing models, e.g., distinct geological models with diverse boundary conditions, see Eq. 2. If available knowledge is insufficient to determine the best model, different models $m$ can be considered, and the respective overall output probability distribution for, e.g., uncorrelated models is computed as (Betz, 2017; Juang et al., 2019):

$$f(y|x,\boldsymbol{\Theta},\boldsymbol{D},\boldsymbol{M}) = \sum f(y|x,\theta,d,m)\omega(m|\boldsymbol{D},\boldsymbol{M}) \qquad (6)$$

$$f(y|x,\theta,d,m) = \int f(y|x,\theta,m)\pi(\theta|d,m)d\theta \qquad (7)$$

In Eq. (6), $\omega(m|\boldsymbol{D},\boldsymbol{M})$ is another distribution weighing each model $m$ in $\boldsymbol{M}$.

The various models $\boldsymbol{M}$, their inputs $\boldsymbol{X}$, parameters $\boldsymbol{\Theta}$, and outputs $\boldsymbol{Y}$, experimental data $d$ can be coupled all together through a Bayesian network, as has been suggested by Sankararaman and Mahadevan (2015) or Betz (2017). One possible configuration of a network coupling elements in $\boldsymbol{M}$, $\boldsymbol{X}$, $\boldsymbol{\Theta}$, $\boldsymbol{Y}$ is illustrated in Figure 1.

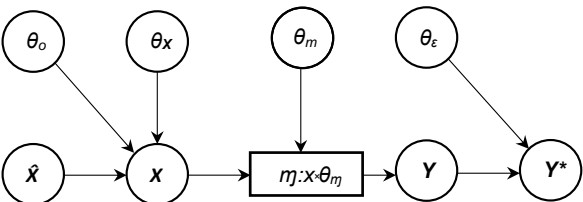

**Figure 1: A configuration of a network coupling some elements in $\boldsymbol{M}, \boldsymbol{X}, \boldsymbol{\Theta}, \boldsymbol{Y}, \boldsymbol{Y}^*$**

The previous description of a general approach to UQ considers uncertainty as that reflected in the analysts' incomplete knowledge about quantities or events. In UQ, to measure or describe uncertainty, subjective probabilities can be used and constrained using historical observations $d$. It is also explicitly shown that model outputs are conditional on historical observations $d$ made available and models $\boldsymbol{M}$ chosen by analysts including the selection of several parameters $\boldsymbol{\Theta}$ and initial and boundary conditions, $BC_m$ and $IC_m$. Based on the above

description, in the following, we analyse some of the challenges that arise when conducting UQ.

As mentioned, back-analysis methods help constrain some elements in $\boldsymbol{\Theta}$; however, given the considerable number of parameters (see Eq. 1-3) and data scarcity, constraining $\boldsymbol{\Theta}$ is often only achieved in a limited fashion. Back-analysis is further challenged by the potential dependency among $\boldsymbol{\Theta}$ or $\boldsymbol{M}$ and between $\boldsymbol{\Theta}$ and $SG_m$, $BC_m$, $IC_m$. We also note that, back analysis, or more specifically, inverse analysis, faces problems regarding non-identifiability,

non-uniqueness, and instability. Non-identifiability occurs when some parameters do not drive changes on the inferred quantities. Non-uniqueness arises because there may be more than one set of fitted or updated parameters that adequately reproduce observations. Instability in the solution arises from errors in observations and the non-linearity of models (Carrera and Neuman, 1986). Alternatively, in specifying, e.g., a joint distribution $f(x,\theta)$ to be sampled, analysts may consider the use of, e.g., Bayesian networks (Albert, Callies, and von Toussaint 2022).

However, under the usual circumstance of a lack of information, establishing such a joint distribution is challenging and requires, in many instances, that analysts encode many additional assumptions (e.g., prior distributions, likelihood functions, independence, linear relationships, normality, stationarity of the quantities and





parameters considered), see e.g., Tang, Wang, and Li (2020); Sun et al. (2021b); Albert, Callies, and von Toussaint (2022); Pheulpin, Bertrand, and Bacchi (2022). A more conventional choice is that $x$ or $\theta$ are *specified* using the

maximum entropy principle in an attempt to specify the least *biased* distributions possible on the given information (Jaynes, 1957). Such distributions are subject to the system's physical constraints based on some available data. The information entropy of a probability distribution measures the amount of information contained in the distribution. The larger the entropy, the less information is provided by the distribution. Thus, by maximising the entropy over a suitable set of probability distributions, one finds the distribution that is least informative in the

sense that it contains the least amount of information consistent with the system's constraints. Note that a distribution is sought over all the candidate distributions subject to a set of constraints. This principle has been questioned, since its validity and usefulness lie in the proper *choice* of physical constraints (Jaynes, 1957; Yano 2019). Doubts are also raised regarding the potential information loss when using the principle. Analysts usually strive in making use of all knowledge available and avoiding unjustified information loss (Christakos, 1990; Flage,

Aven, and Berner, 2018).

Options to address the parametrisation challenge also include surrogate models, parameters reduction, and model learning (e.g., Lu and Lermusiaux, 2021; Sun et al., 2021b; Albert, Callies, and von Toussaint, 2022; Degen et al. 2022; Liu et al., 2022). Surrogate models are learnt in order to replace a complicated model with an inexpensive and fast approximation. Parameters reduction is achieved based on either principal component analysis or global

sensitivity analysis to determine which parameters significantly impact model outputs and are essential to the analysis (Degen et al., 2022; Wagener, Reinecke, and Pianosi, 2022). Remarkably, versions of the model learning option do not need any prior information about model equations $E_{\mathbb{m}}$ but require local verification of conservation laws in the data $d$ (Lu and Lermusiaux, 2021). These approaches still require large past data sets, sourced systematically, which is a frequent limitation in geohazard assessments. More importantly, however is that, like

many models, even those exhaustively validated, the *credibility* of unobserved surrogate model outputs can always be questioned, since, for instance, records may miss crucial events or models may fail to reproduce outputs caused by recorded abrupt changes (e.g., extreme velocities of turbidity currents) (Alley, 2004; Woo, 2019). An additional point is the issue of incomplete model response, which refers to a model not having a solution for some combinations of the input variables (Cardenas, 2019; van den Eijnden, Schweckendiek, and Hicks, 2021).

In bypassing the described challenges when quantifying uncertainty, simplifications are usually enforced, sometimes unjustifiably, in the form of assumptions, denoted here by the set $A$, and this set can include one or more of the assumptions listed in Table 1. Note that the set of assumptions can be increased with those assumptions imposed by the use of specific models $M$ (e.g., conservation of energy, momentum or mass, Mohr-Coulomb's failure criterion).

**Table 1.** Some enforced assumptions in UQ for geohazard assessments

| |
|---|
| Predictions (non-observed outputs) of $Y$ are credible, despite models only reproducing responses based on historical data $d=\{\hat{Y}=\hat{y}, X=\hat{x}\}$, $d \in D$. |
| A model has a solution for any combination of the input quantities $X$. |
| Elements in $X$ are fully specified. |
| Elements in $X$ are mutually independent. |
| The joint distribution $f(x,\theta)$ distributes according to the maximum entropy principle. |
| If measurements are available, some input quantities $X$ are set to specific values $x=\hat{x}$. |
| Input quantities $X$ are set to constant values $x_0$, that is $X=x_0$. |
| Some $\theta$ are set to specific point values and are mutually independent. |
| Some $\theta$ are independent of $SG_{\mathbb{m}}$, $BC_{\mathbb{m}}$, $IC_{\mathbb{m}}$. |
| $SG_{\mathbb{m}}$, $BC_{\mathbb{m}}$, $IC_{\mathbb{m}}$ are set to be constant. |
| When some data $d$ is available in the form of measurements $\{\hat{y}, \hat{x}\}$, likelihood functions $\mathcal{L}[\mathbb{m}(\theta\|d)]$ are mutually independent. |





## 3 Addressing the challenges in uncertainty quantification


From the previous section, we saw that it is very difficult in geohazard assessments to meet data requirements for the *ideal* parameterisation of models. Further, we have noted that, although fully parameterised models could potentially be accurate at reproducing data from past events, these may turn out to be inadequate for unobserved outputs. We also made explicit that predictions are not only conditional on $\Theta$ (including priors, likelihood functions and linked

hyperparameters) but possibly also on $SG_{\eta}, BC_{\eta}, IC_{\eta}$, see Eq. (1-7). Ultimately, assumptions made also condition model outputs. More importantly, note that when only some model quantities or parameters can be constrained by data *d*, the modelling will only reflect some aspects of the uncertainty involved. If the above challenges remain unaddressed, UQ lacks credibility. To address such challenges and provide increased credibility, clarifications and reinterpretation of some fundamental concepts together with practical simplifications may be required, and these

are discussed in the following. Table 2 shows the major challenges found and how they are addressed in related literature, while in Table 3 some clarifications or considerations put forward by us are displayed. The discussion in this section builds on previous analysis by Aven and Pörn (1998), Apeland, Aven, and Nilsen (2002), Aven and Kvaløy (2002), Nilsen and Aven (2003), Aven and Zio (2013), Khorsandi and Aven (2017), and Aven (2019).

**Table 2. Major challenges and options to address them in geohazard assessments**

| Challenges, CH | Options to address the challenges, O |
|---|---|
| ***Challenges related to the model outputs and system responses*** | |
| ***CH1***. *Predictions* of *y* lack credibility since these are model outputs not recorded in the data *d*, <br> ***CH2***. A model does *not have a solution* for a feasible *combination of the input quantities **X***. | ***O1***. *Credibility* of predictions is *judged* in terms of physical consistency checks (Wagener, Reinecke, and Pianosi, 2022) and by examining the ability of models to reproduce disruptive changes recorded in the data (Alley, 2004). <br> ***O2***. Predictions by *Bayesian forecasting methods*. Based on a prior distribution for *y*, a posterior distribution of *y* is obtained by including the information provided by the model prediction in the form of model likelihood (Montanari and Koutsoyiannis, 2012). |
| ***Challenges related to input quantities*** | |
| ***CH3***. Data available *d* may *not include all the historic crucial events or disruptive changes*, <br> ***CH4***. Some input quantities ***X*** remain *unknown (unidentified)* to analysts during an assessment, <br> ***CH5***. *The distribution f(x) or the bounds* of *x* are *unknown*, <br> ***CH6***. Some input quantities ***X*** may be mutually *dependent*. | ***O3***. Using the *maximum entropy principle* to specify the distributions based on the *choice* of constraints regarding physics of the phenomena involved. Constraining the distributions by data including data other than measurements to reduce unjustified information loss (Jaynes, 1957; Christakos, 1990; Betz, 2017; Yano 2019). <br> ***O4***. *Counterfactual analysis* in which alternative events to observed facts including disruptive changes are '*imagined*', *assumed,* and explored to obtain alternative system's responses using models (Pearl, 1993; Woo, 2019). <br> ***O5***. Exhaustive investigation of input uncertainty using the *assumptions deviation approach* to specify input distributions (Aven, 2013). |
| ***Challenges related to the parameters and models*** | |
| ***CH7***. The *distribution f(θ) or the bounds* of *θ* are *unknown*, <br> ***CH8***. Some *θ* may be *dependent* on ***SG**_{\eta}, **BC**_{\eta},* or ***IC**_{\eta}*, <br> ***CH9***. *Likelihood functions $\mathscr{L}[\eta(\theta|d)]$* may be mutually *dependent*, <br> ***CH10***. *Models $\eta$ in $\mathcal{M}$* may be mutually *correlated*. | ***O3***. Using the *maximum entropy principle* as described above. <br> ***O6***. A joint distribution of ***Θ**, **SG**_{\eta}, **BC**_{\eta}, **IC**_{\eta}, **X*** for each *$\eta$*, can be specified by encoding other assumptions (e.g., prior distributions, likelihood functions, independence, linear relationships, normality, stationarity) in *Bayesian networks* (Albert, Callies, and von Toussaint, 2022). <br> ***O7***. Using *surrogate models*, *parameters reduction*, and *model learning* (Lu and Lermusiaux, 2021; Albert, Callies, and von Toussaint, 2022). |


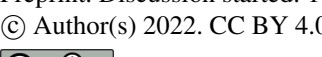



**Table 3. Some clarifications and considerations to address the challenges in UQ**

*C1*. *Uncertainty* reflects that analysts' knowledge *about quantities or events is incomplete*.

*C2*. Models are simplifications, *mainly used for understanding* the performance of the system and *approximating its responses*; they are part of the knowledge of the system, and they do not introduce uncertainty.

*C3*. The focus is on the quantification of the *uncertainty of the system responses* rather than on the accuracy of a model reproducing recorded data.

*C4*. Predictions are *conditional* on the model(s) chosen together with a number of *assumptions* made by analysts.

*C5*. The *specification* of the joint distribution *f(x,θ) cannot solely rely on the use of maximum entropy principle*, but on the full scrutiny of background knowledge $K$, and such a distribution is better specified *using knowledge-based probabilities*.

*C6*. Some elements in the parameter set $\Theta$ *are not quantities* or properties of the system as such, and there could not be uncertainty about them.

*C7*. Analysts may *choose* a model or a set of them, which are *believed* or *judged* to be the *best credible models*.

Among the clarifications, we consider a major conceptualisation suggested by the literature, which is the definition of uncertainty. Uncertainty refers to incomplete information or knowledge about a hypothesis, quantity, or the

occurrence of an event (Society for Risk Analysis, 2018). In Table 3, we denote this clarification as *C1*. Embracing this definition has some implications for uncertainty quantification using geohazard models. We make use of these implications to address the major complications and challenges found. For instance, if uncertainty is measured in terms of probability, one such implication is that analysts are discouraged to use the so-called frequentist probabilities due to the fact these probabilities do not measure uncertainty or lack of knowledge. Rather such

probabilities reflect frequency ratios representing fluctuations or variation in the outcomes of quantities. Frequentist probabilities are of limited use because these assume that quantities vary in large populations of identical settings, a condition which can hardly be justified for rather few geohazard quantities due to both the often one-off nature of many geohazard features and the impossibility to verify or validate data by, e.g., a large number of repeated tests. Thus, considering the usual constraints in data, as well as the nature of geohazard events,

a more meaningful and practical approach can be suggested to actually measure uncertainty, namely the use of knowledge-based (also referred to as judgemental or subjective) probabilities (Aven 2019). A knowledge-based probability is understood as an expression of the degree of belief in the occurrence of an event or quantity by a person assigning the probability and conditional on the available knowledge $K$. Such knowledge $K$ includes not only data in the form of measurements $d$ made available to the analysts at the time of the assessment, but also other

data sources in $D$ together with the models $M$ chosen by the analysts for the prediction, as well as the modelling assumptions $A$ made by those analysts. Accordingly, we have in total that, to describe uncertainty about, e.g., $x$ or $y$, probabilities are assigned based on $K$ and, therefore, we acknowledge that those probabilities are conditional on $K$. In the previous section, we have made explicit the conditional nature of uncertainty on measured data $d \in D$ and models $M$ and wrote, explicitly, for the overall output probability distribution, the expression $f(y|x,\Theta,D,M)$,

(see Eq. 6). If assumptions $A$ are also acknowledged as conditioning uncertainty, we write more explicitly $f(y|x,\Theta,D,M,A)$ or equivalently $f(y|x,\Theta,K)$. We can therefore write:

$$f(y|x,\Theta,K) = f(y|x,\Theta,D,M,A) \tag{8}$$

The meaning of this expression can be elucidated next. If, in a specific case, we would write $f(y|x,\theta,D)$, it means that $D$ summarises all the knowledge that analysts have to calculate $y$ given (realised or known) $x$ and $\theta$. The full





expression in Eq. 8 implies, accordingly, that to calculate $y$, and when knowing $x$ and $\theta$, the background knowledge includes $\mathcal{D}$, $\mathcal{M}$, $\mathcal{A}$. Note that $K$ can also be formed by observations, justifications, rationales, and arguments, thus Eq. 8 can be further detailed to include these aspects of $K$. Structured methods exist to assign knowledge-based probabilities (see, e.g., Apeland, Aven, and Nilsen, 2002; Aven 2019). Here we should note, however, that since models form part of the available background knowledge $K$, models can also inform these knowledge-based

probability assignments. It follows that, based on knowledge-based input probabilities, an overall output probability distribution calculated using models is also of subjective character or knowledge based, although uncertainty quantification using sampling could provide some frequency resemblance or interpretation (Jaynes, 1957). Some of the implications of using knowledge-based probabilities are described throughout this section.

     According to the left column in Table 2, the focus of the challenges relates to the model outputs, more specifically

predictions (*CH1* and *CH2*), input quantities (*CH3-CH6*), parameters (*CH7-CH9*), and models (*CH10*). We recall that as defined, uncertainty quantification helps determine the system's response uncertainty based on specified input quantities using models. Accordingly, the focus of an assessment is on the potential system's responses. When it comes to quantifying uncertainty, the focus is often on uncertainty about future non-observed responses $Y^*$, which are approximated by the model output $Y$ considering some specified input quantities $X$. We recall that

then $Y^*$ and $X^*$ are quantities that are unknown at the time of the analysis, but will, if the system being analysed is implemented, take some value in the future, and possibly become known. Thus, during an assessment $Y^*$ and $X^*$ are the uncertain quantities of the system since we have incomplete knowledge about $Y^*$ and $X^*$. Based on these grounds, the output-prediction error $\varepsilon$ (described in the previous section) being the mismatch between the model prediction values $y$ and non-observed system's responses values $y^*$, usually suggested to correct the

imperfect model output $y$, can only be specified on the basis on the scrutiny of $K$.

     Another consequence of considering the definition of uncertainty put forward, which solely links uncertainty to quantities or events, and taking into account that models are only mathematical artefacts, is that models, as such, are not to be linked to uncertainty. Models, per se, do not introduce uncertainty, but they are likely to be inaccurate. Accordingly, another major distinction is to be set in place. We recall that models, by definition, are

simplifications, approximations of the system being analysed; they express or are part of the knowledge of the system and should therefore be solely used for understanding the performance of the system rather than for illusory perfect predictions. In Table 3, we denote the latter clarification as *C2*.

     Regarding the challenges *CH1* and *CH2*, we should note that geohazard analysts are often more interested in predictions rather than known system outputs. For instance, predictions are usually required to be calculated for

input values that are not contained in the validation data. We consider that predictions are those model outputs not observed or recorded in the data, i.e., extrapolations out of the range of values covered by observations (e.g., mudflow extreme velocities). Thus, the focus is on the quantification of the uncertainty of extreme system's responses rather than on the accuracy of a model reproducing recorded data. This is the clarification *C3* in Table 3. Considering this, models are yet to provide accuracy in reproducing observed outputs but, more importantly,

afford credibility in predictions. Such credibility is to be assessed, mainly, in terms of judgements since conventional validation cannot be conducted using non-observed outputs. Recall that model accuracy usually relates to the comparison of model outputs with experimental measurements (Roy and Oberkampf, 2011; Aven and Zio, 2013) and is the basis for validating models. Regarding credibility of predictions, Wagener, Reinecke, and Pianosi (2022) have reported that such credibility can be mainly judged in terms of the physical consistency





260 of the predictions, by checks rejecting physically impossible representations of the system. The credibility of predictions may also include the verification of the ability of models to accurately reproduce disruptive changes recorded in the data (Alley, 2004). However, as we have made explicit in the previous section, models' predictions are conditional on a considerable number of critical assumptions and choices made by analysts (see Table 1 and clarification C4 in Table 3). Therefore, predictions can only be as good as the quality of the assumptions made.

265 The assumptions could be wrong, and the examination of the impact of these deviations on the predictions must be assessed. To provide credibility of predictions, such assumptions and choices should be justified and scrutinised, ref. option *O5* in Table 2. The option *O5* addresses the challenge *CH1*; however, when conducting UQ using models, *O5* has a major role when investigating input uncertainty, which is discussed next.

A critical task in uncertainty quantification is the quantification of input uncertainty. As shown in Table 2, such

270 uncertainty is not only to be associated with the lack of knowledge of the distribution $f(x)$ or bounds of $x$, *CH5*, and some other sources are to be considered. For example, input uncertainty may originate from the situation that historic crucial events or disruptive changes are missing in the records, *CH3*, or from the condition in which critical quantities in $X$ may remain unidentified to analysts during an assessment, *CH4*. We recall that analysts can unintendedly fail to identify relevant elements in $X$ due to insufficiencies in data or limitations of existing models.

275 For example, during many assessments, trigger factors that could bring a soil mass to failure could remain unknown to analysts (e.g., Hunt et al. 2013; Clare et al. 2016; Leynaud et al. 2017; Casalbore et al. 2020). Uncertainty quantification based on models requires simulating sampled values from $X$, and elements in $X$ can possibly be mutually dependent; however, the joint distribution of $X$, namely $f(x)$, is often also unknown. This is the challenge *CH6*. Considering the potential challenges *CH3* to *CH6*, to specify $f(x)$ we cannot solely rely on the use of the

280 maximum entropy principle, as described in the previous section, since it may fail to advance an exhausted uncertainty quantification in the input, e.g., by missing relevant values not recorded in the measured data. This would undermine the quality of predictions and therefore uncertainty quantification. Recall that the principle suggests the use of the least informative distribution among candidate distributions constrained solely on measurements. Using counterfactual analysis, as described in Table 2, is an option, but this will also fail in

285 providing quality in predictions, since this analysis focuses on the analysis of counterfactuals (a part of the data $D$) and little on the overall knowledge available $K$. Note that system's knowledge $K$ includes, e.g., among others, the assumptions made in the quantification of uncertainty, such as those shown in Table 1. Further note that such assumptions not only relate to data, but also to input quantities, modelling, and predictions. Thus, it appears that the examination of these assumptions should, desirably, be at the core in quantifying uncertainty in geohazard

290 assessments, as suggested in Table 2, option *O5*. The assessment of deviations of assumptions has been suggested originally by Aven (2013) and Khorsandi and Aven (2017). An assumption deviation risk assessment evaluates different deviations, their associated probabilities of occurrence, and the effect of the deviations. The major distinctive features of the assumption deviation risk assessment approach are the evaluation of the credibility of the knowledge $K$ supporting the assumptions made, together with the questioning of the justifications supporting

295 the potential for deviations. The examination of $K$ can be achieved by assessing the justifications for the assumptions made, the amount and relevance of data or information, the degree of agreement among experts, and the extent to which the phenomena involved are understood and can be modelled accurately. Note that justifications might be in the form of direct evidence becoming available, indirect evidence from other observable quantities, supported by modelling results, or possibly inferred by assessments of deviations of assumptions. This approach



is succinctly demonstrated in the following section. Accordingly, we suggest specifying $f(x)$ in terms of knowledge-based probabilities in conjunction with the investigation of input uncertainty using the assumptions deviation approach. This is identified as consideration $C5$ in Table 3.

Another point to consider is that, when uncertainty reflects analysts' (lack of) knowledge about quantities or events and is measured in terms knowledge-based probabilities, analysts should be aware that conditionality among

elements in $X$ only implies that increased knowledge about, e.g., a quantity $X_1$ will change the uncertainty about another quantity $X_2$, if $X_2$ is conditional on $X_1$. The expression that denotes this is conventionally written as $X_2|X_1$. This interpretation may be exploited by analysts when specifying, e.g., the joint distribution $f(x,\theta)$. Analysts, for example, may simplify the analysis when, according to the scrutiny of $K$, increased knowledge about, e.g., $X_1$ will not result in increased knowledge about another quantity, e.g., $X_2$, and, for example, if a distribution $f(y|x_1x_2)$ is to

be specified, we may have then that $f(y|x_1x_2)$ reduces to $f(y|x_1)f(y|x_2)$, according to probability theory. Apeland, Aven, and Nilsen (2002) have illustrated how conditionality in the setting of knowledge-based probabilities can inform the specification of a joint distribution.

The parameterisation problem, which involves the challenges $CH7$ to $CH9$ in Table 2, warrants exhaustive consideration. Addressing these challenges also requires some reinterpretation. To start, we recall that, parameters,

by definition, are coefficients determining specific functions among a family of potential functions modelling the system. Those parameters, eventually, constrain a model's responses. Recall that, $y$ as realisations of $Y$ are the model responses when $X$ takes the values $x$ and some parameters $\theta \in \Theta$ and models $\eta \in M$ are used. Thus, as shown in the previous section, any output $y$ is conditional on $\theta$ and so is the uncertainty attached to $y$. Further note that, taking into consideration the definition of uncertainty previously stated, which attributes uncertainty to events,

quantities, or hypothesis solely, we may say that, unless a parameter is associated with a quantity or property of the system, non-quantity parameters, being merely artefacts in the models, are not to be linked to any uncertainty. This is identified as clarification $C6$ in Table 3. Using the uncertainty definition previously put forward and looking at Eq. 3, analysts may consider that, for instance, parameters $\theta_\varepsilon$, which are linked to the output-prediction error $\varepsilon$, some model parameters in $\Theta_\eta$, the vector associated with observation/measurement errors $\Theta_o$ and the overall

attached hyperparameters linked to probability distributions (including priors, likelihood functions), are not quantities or properties of the system as such; they are modelling artefacts, and therefore, it is questionable to consider uncertainty about them. Thus, focused on the uncertainty of the system responses rather than model inaccuracies, the analysts may consider strictly assigning uncertainty to those parameters that represent physical quantities and fix with single values those that are not quantities as such. To help identify those parameters to

which some uncertainty can be linked, we can scrutinise, e.g., the physical nature of these. Note, however, that in fixing parameters to a single value, we still can make use of back analysis procedures, as mentioned previously. Analysts may have some additional basis to specify parameters values when the background knowledge available $K$ is scrutinised to verify that $K$, including not only data measurements but other sources of data, models, assumptions made, and arguments, strongly supports a specific parameter value. Based on this interpretation,

setting the values of these non-quantity-based parameters to a single value reduces, considerably, the complications in quantifying uncertainty. It also follows that, analysts are encouraged to make explicit that model outputs are conditional on these fixed parameters, as well as on the model or models chosen, as we have shown in the previous section. The latter interpretation also leads us to argue that the focus of the quantification is on the uncertainty of the system response rather than the inaccuracies of the models. This, of course, implies that, in a practical sense



and in the context of geohazard assessments, once a clear differentiation between parameters and input quantities
      is made, and a model or a set of them believed or judged to be the best models providing the most credible
      predictions are chosen, uncertainty quantification can then proceed. This parsimonious modelling approach is
      identified as consideration *C7* in Table 3. This latter consideration addresses, to an extent, the challenge *CH10*.
      In the following section, we further illustrate the above discussion by analysing a documented case in which UQ

in a geohazard assessment was informed by modelling using sampling procedures.

**4 Case analysis**

To further describe the proposed considerations, we analyse a case reported in the specialised literature. The case
deals with the quantification of uncertainty of geological structures, namely uncertainty about the subsurface
stratigraphic configuration. Conditions in the subsurface are highly variable, whereas site investigations only

provide sparse measurements. Consequently, subsurface models are usually inaccurate. At a given location,
      subsurface conditions are unknown until accurately measured. Soil investigation at all locations is usually
      impractical and uneconomical, and point-to-point condition variation cannot therefore be known (Vanmarcke,
      1984). Such uncertainty means significant engineering and environmental risk to, e.g., infrastructure built on the
      surface. One way to quantify this uncertainty is calculating the probability of every possible configuration of the

geological structures (Tacher et al. 2006; Thiele et al. 2016; Pakyuz-Charrier et al. 2019). Sampling procedures
      for UQ are helpful in this undertaking. We use an analysis and information by Zhao et al. (2021), which refer to a
      site located in the Central Business District, Perth, Western Australia, where 6 boreholes were executed. The case
      has been selected taking into account its simplicity to illustrate the points of this paper, but at the same time, it
      provides details to allow some discussion. Figure 2 displays the system being analysed.

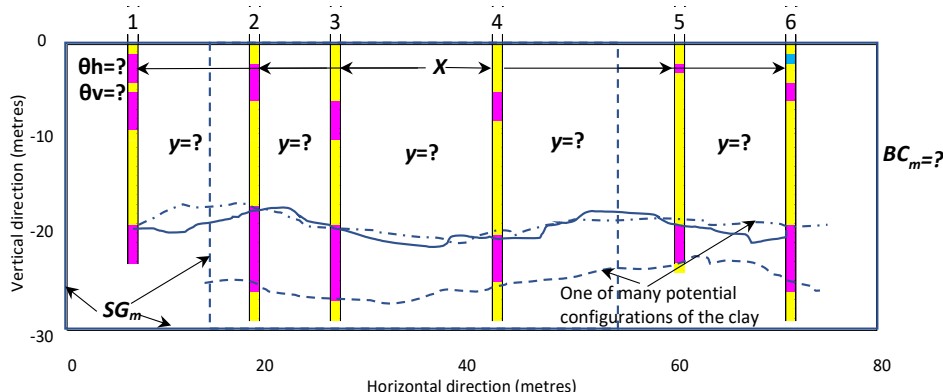

**Figure 2: Borehole logs in colours and longitudinal section reported by Zhao et al. (2021) located in the Central Business District, Perth, Western Australia. The records correspond to information of the six boreholes. Three types of materials are revealed by the boreholes including sand (yellow), clay (magenta), and gravel (blue).**

In the system under consideration, a particular material type to be found in a non-bored point, a portion of terrain
      that is not penetrated during soil investigation, is unknown and thus uncertain. The goal is, therefore, the
      computation of the probability of encountering a given type of soil in these points. In Zhao et al. (2021) the focus
      is on calculating the probabilities of encountering clay in the subsurface, and the approach advocated was a
      sampling procedure to generate many plausible configurations of the geological structures and evaluating their

probabilities based on their frequencies. To calculate the probability of encountering a given type of soil $c$, $p(y=c)$,
in a non-penetrated point in the ground, Zhao et al., (2021) used a function that depends on two correlation
parameters, namely the horizontal and vertical scale of fluctuation $\theta_h$ and $\theta_v$. Note that spatial processes and their
properties are conventionally assumed as spatially correlated. Such spatial variation may presumably be
characterised by correlation functions, which depend on a scale of fluctuation parameter. The scale of fluctuation

measures the distance within which points are significantly correlated (Vanmarcke, 1984). Eq. 9 describes the
basic components of the model chosen by Zhao et al., (2021) (specific details are given in the Appendix to this
paper):

$$\eta: X_{s_x} \times \Theta_\eta \rightarrow Y_s \rightarrow p(y=c) \qquad (9)$$

where $X$ is the collection of all quantities at borehole-points $s_x$ which can take values $x$ from the set {sand, clay,

gravel}, according to the setting in Figure 2. $Y$ is the collection of all quantities with values $y$ at non-borehole
points $s_y$. The values $y$ together with the values $x$ are sampled and probabilities computed on the basis of a chosen
model using the parameters $\theta_h$=11,1 and $\theta_v$=4,1 metres, $\theta_h$, $\theta_v \in \Theta_\eta$. Using the maximum likelihood method, the
parameters were determined based on the borehole data revealed at the site. In the determination of parameters,
the sampling of uniform and mutually independent distributions of $\theta_h$ and $\theta_v$ was the procedure advocated. The

system is further described by a set of equations $E_\eta$ (a correlation function and a probability function), the spatial
domain geometry $SG_\eta$ (a terrain block of 30 x 80 metres), and the boundary conditions $BC_\eta$ (the conditions at the
borders). More details are given in the Appendix to this paper. Since this system is not considered time dependent,
the initial conditions $IC_\eta$ were not specified.

The summary results reported by Zhao et al. (2021) are shown in Figure 3. In Figure 3, the most probable

stratigraphic configuration along with spatial distribution of the probability of the existence of clay is displayed.
The authors focused their attention on this sensitive material, which likely represents a risk to infrastructure built
on the surface.

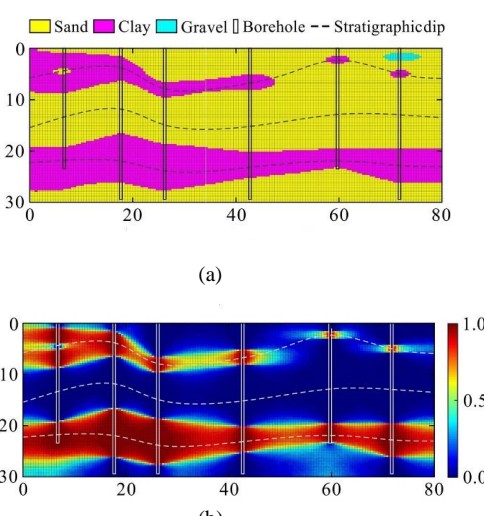

**Figure 3: Zhao et al. (2021) findings shown in their Figure 9. (a) Most probable stratigraphic configuration. (b) Spatial distribution of the probability of the existence of clay. Authorisation of reproduction: License number 5351891245835. Distances in metres.**





Zhao et al., (2021) stated that "*characterisation results of the stratigraphic configuration and its uncertainty are*
*consistent with the intuition and the state of knowledge on site characterisation*". Next, throughout Zhao et al.'s
(2021) analysis the following assumptions were enforced (Table 4), although these were not explicitly disclosed
by the authors.

**Table 4. Assumptions enforced by Zhao et al., (2021)**

Predictions (non-observed outputs) are credible.
Likelihood functions $\mathcal{L}[\eta(\theta|d)]$ were set to be mutually independent.
For the determination of parameters and model, $f(\theta_h, \theta_v)$ distributes according to the maximum entropy principle
and $\theta_h$ and $\theta_v$ are mutually independent.
Specified elements $X$ are complete.
Input quantities $X$ were set to the measured values, i.e., $x = \hat{x}$ (no inaccuracies in data).
$\theta$ are independent of $SG_\eta$ and $BC_\eta$.
$SG_\eta$, $BC_\eta$ were set to be constant.

Unfortunately, the authors did not report enough details on how the majority of these assumptions are justified.
We, however, should note that providing these justifications was not the objective of their research. Yet, we analyse
here how these can be justified by both scrutinising the supportive knowledge $K$ and using some elements of the
assumption deviation approach, described in the previous section. Table 5 summarises the analysis conducted and
only reflects the most relevant observations and reservations identified by us. Accordingly, the information in
Table 5 may not be exhaustive but still useful for the desired illustration. Table 5 displays some observations by
us related to the credibility of the knowledge $K$. The examination of $K$ is achieved by assessing the amount and
relevance of data or information, the extent to which the phenomena involved are understood and can be modelled
accurately, the degree of agreement among experts, and the justifications for the assumptions made. Observations
regarding the justifications for potential deviations of assumptions also form the analysis.

Not surprisingly, the observations in our analysis are concentrated on the predictions' credibility. Recall that the
focus of UQ is on the system's response, which is approximated by model predictions (considerations *C2* and *C3*
in Table 3). We note, for example, that, although the use of correlations is an accepted practice and a practical
simplification, correlation functions appear to be counterintuitive to model geological structures or domains and
do not help much in understanding the system (consideration *C2* in Table 3). Recall that such structures are mainly
disjoint domains linked to a finite set of possible categorical (masses of soil or rock) rather than continuous
quantities. Next, the variation of such structures can occur by abrupt changes of materials, thus the use of, for
example, smoothed functions, as correlation functions may be, to represent them requires additional consideration.
Further, the physical basis of the correlation functions is not clear and physical models based on deposition
processes may be suggested (e.g., Catuneanu et al., 2019). We should further note that a potential justification for
the deviation of the assumption regarding the credibility of predictions is that knowledge from additional sources
such as surface geology, sedimentology, local geomorphic setting, and structural geology was not explicitly taken
into account in the quantification of uncertainty. The revision of this knowledge can contribute to reduce the
probability of the deviation in predictions. Based on the observations in Table 5, we can conclude that there is
potential to improve the credibility of predictions.






**Table 5. Examination of supporting knowledge K and justifications for the potential deviation of assumptions (part a)**

| The amount and relevance of data or information | The extent to which the phenomena involved are understood, and accurate models exist | The degree of agreement among experts | Justifications for the assumptions made | Justifications supporting the potential deviations |
|---|---|---|---|---|
| *Assumption:* Predictions of *Y* are credible | | | | |
| The analysis is only based on borehole information; however, such investigation is exceptionally exhaustive, 6 boreholes | The physical basis for using correlations is dubious and models based on the deposition process can be considered<br><br>Variation of geological structures can occur by abrupt changes, thus the use of smoothed functions to represent them requires additional consideration<br><br>Global rather than local correlation between spatial quantities has been used, possibly misrepresenting geological structures variation | The use of correlations is an accepted practice in the field | Exhaustive borehole information | The knowledge of surface geology, sedimentology, local geomorphic setting, and structural geology was not explicitly incorporated into UQ |
| *Assumption:* Likelihood functions $\mathcal{L}[m(\theta\|d)]$ were set to be mutually independent | | | | |
| *Assumption:* $f(\theta_h,\theta_v)$ distributes according to the maximum entropy principle and $\theta_h$ and $\theta_v$ are set to be mutually independent | | | | |
| Data of the six boreholes has been used to calibrate the model chosen; however, knowledge of surface geology, local sedimentology, geologic/geomorphic setting, and structural geology was not explicitly incorporated into the analysis | | Based on the maximum likelihood method, a model judged to be the best model was chosen | Dependency between $\theta_h$ and $\theta_v$, cannot be supported by general knowledge and such dependency hardly can be enforced<br><br>A joint distribution $f(x,\theta,sg_m,bc_m)$ could have been investigated when determining the models and parameters, but establishing such a joint distribution is challenging and requires, in many instances, that analysts encode many additional assumptions, which hardly can be justified | An increased revision of *K*, could have been useful to specify $f(x)$ and $f(\theta)$ providing a richer information than that suggested by the maximum entropy principle, regarding how *X* or *Θ* take values |





**Table 5.** Examination of supporting knowledge 𝒦 and justifications for the potential deviation of assumptions (part b)

| The amount and relevance of data or information | The extent to which the phenomena involved are understood, and accurate models exist | The degree of agreement among experts | Justifications for the assumptions made | Justifications supporting the potential deviations |
|---|---|---|---|---|
| *Assumption:* Specified elements $X$ are complete | | | | |
| | | Input quantities were considered fully specified | | Another type of material was revealed by other soundings in the area, i.e., silt (see sources in Zhao et al. 2021), and depending on the revision of $K$, this fourth material could be considered. |
| *Assumption:* Input quantities $X$ were set to measured values $x=\hat{x}$ | | | | |
| | Errors during surveys may have resulted in horizontal positioning inaccuracies | Data was judged to be accurate | There is usually not data basis to calculate these errors | |



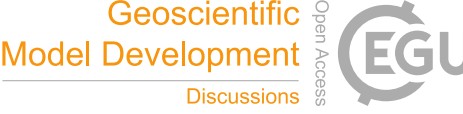

The choices made by Zhao et al., (2021) regarding the use of parameters with fixed values together with the choice
for a single best model can be highlighted and illustrate the points raised in the considerations *C6* and *C7* (Table
3)*, respectively. The maximum likelihood method supported these choices, a back analysis method focused on the
matching of measurements and calculated model outputs using different assumed values for $\theta_h$ and $\theta_v$. We highlight
that, a model judged to be the best model was chosen. This includes the specification of a particular spatial domain

geometry $SG_{\eta}$. Investigating the impact of variation of $SG_{\eta}$ was considered unnecessary. There was no need to
specify several competing models, which is line with our consideration labelled as *C7* in this paper.

Although Zhao et al. (2021) investigated the joint distribution *f(x)*, which was sampled to calculate probabilities,
someone can suggest that the joint distribution $f(x,\theta,sg_{\eta},bc_{\eta})$ could have been produced when determining the
models and parameters for calculating probabilities. Nevertheless, we can argue that establishing such a joint

distribution is challenging and requires, in many instances, that analysts encode many additional assumptions (e.g.,
prior distributions, likelihood functions, independence, linear relationships, normality, stationarity of the quantities
and parameters considered).

A more crucial reservation derived from the analysis of potential deviations of assumptions, which might impact
considerably the credibility of predictions, comes from revisiting the knowledge sources of Zhao et al.'s (2021)

analysis, available from https://australiangeomechanics.org/downloads/. Another type of sensitive material was
revealed by other soundings in the area, more specifically, silt. Depending on the revision of $\mathit{K}$, this fourth
suspected material could be analysed in an extended uncertainty quantification of the system. Note that, originally,
the input quantities *X* were assumed to take values *x* from the set {*sand*, *clay*, *gravel*}. Such an assumption was
based on the records of six boreholes which were believed accurate. The latter illustrates the relevance of the

consideration *C5* in Table 3.

Another interesting choice made by Zhao et al. (2021) is that they disregarded the possibility of incorporating into
the modelling measurement errors in the borehole data, probably because this data was judged to be accurate. We
recall in this respect that these errors reflect the inaccuracy of the ground model rather than the uncertainty about
the system. We further note that, as stated for consideration *C6* (Table 3), we can hardly justify attaching

uncertainty to measurement error parameters, since measurement errors are not a property of the system. The same
can be said for the parameters $\theta_h$ and $\theta_v$, which are not quantities of the system. Note that their physical basis is
questioned. We should note, however, that assuming coefficients for the parameters $\theta_h$ and $\theta_v$ is an established
practice (Vanmarcke, 1984, Lloret-Cabot et al., 2014, Juang et al., 2019). We also point out that uncertainty
quantification in this kind of systems is to an extent sensitive to the choice of scale of fluctuation values

(Vanmarcke, 1984), and that the use of a global rather than local correlation between spatial quantities can
potentially misrepresent geological structures variation. Accordingly, a further examination of the existing $\mathit{K}$ can
justify undertaking an assessment of the impact of a deviation of assuming a local rather than global scale of
fluctuation.

Overall, the Zhao et al. (2021) analysis is to an extent based on the previously suggested definition of uncertainty,

ref. the consideration *C1* in Table 3.

We should stress that the Zhao et al. (2021) uncertainty quantification refers specifically to the ground model
described at the beginning of this section. In other words, the probabilities displayed in Figure 3b are conditional
on the parameters chosen ($\theta_h$=11,1 and $\theta_v$=4,1 metres); the model selected (described by the Eq. 9, A-1 and A-2 in
the Appendix to this paper); the specified spatial domain geometry $SG_{\eta}$ (a terrain block of 30 x 80 metres); and



ultimately the assumptions made (listed in Table 4). This information is to be reported explicitly to the users of the results. This reflects the clarification *C4* in Table 3.

Regarding the consideration of subjective probabilities, there has been agreement, to an extent, on their use in this kind of UQ since Vanmarcke (1984). However, the use of knowledge-based probabilities in the extension described here is recommended given the illustrated implications to advance UQ (as discussed in the previous section and stated in consideration *C5*). For example, increased examination of $K$, might have resulted in using a more informative distribution $f(\theta_h,\theta_v)$ other than the uniform distribution, and in turn, different values for $\theta_h$ and $\theta_v$, as well as a different model. Recall that the selection of the model and determination of parameters were based on the maximum likelihood method, which only makes use of measured data $d$. Note, however, that when Zhao et al. (2021) calculated the probabilities, they sampled an improved joint distribution $f(x)$ using the parameters $\theta_h$ and $\theta_v$ and a chosen model.

In our analysis of Zhao's et al. (2021) assessment, the examination of supporting knowledge $K$ resulted essentially in:

(i)    judging the credibility of predictions;

(ii)   providing justifications for undertaking an assessment of assumption deviations considering, e.g., further analysis involving the modelling of a fourth material;

(iii)  considering additional data other than the borehole records, such as surface geology, sedimentology, local geomorphic setting, and structural geology;

(iv)   analysing the possibility of distinct geological models with diverse spatial domain geometry and local correlations; and

(v)    ultimately, further examining the existing $K$.

**5 Conclusions**

In this paper, we have discussed challenges in uncertainty quantification, UQ, based on models, for geohazard assessments. Beyond the parameterisation problem, the challenges include how to assess the quality of predictions required in the assessments, the quantification of uncertainty in the input quantities, the consideration of the impact of choices and assumptions made by analysts. Such challenges arise from the common-place situation of limited data and the one-off nature of geohazard features. If these challenges are kept unaddressed, UQ lacks credibility. Here, we have formulated seven considerations that may contribute to providing increased credibility in the quantifications. For example, we proposed understanding uncertainty as lack of knowledge, a condition that can only be attributed to quantities or events in the system under consideration. Another consideration is that the focus of the quantification should be more on the uncertainty of the system response rather than the accuracy of the models used in the quantification. We drew attention to the clarification that models, in geohazard assessments, are simplifications used for predictions approximating the system's responses. We have also considered that since uncertainty is only to be linked to properties of the system, models, as such, do not introduce uncertainty. Inaccurate models can, however, produce poor predictions, and under these circumstances, the uncertainty about system response is to be judged as large. Such models should be rejected, and increased examination of background knowledge will be required to credibly quantify uncertainty.  We also put forward that there could not be uncertainty about those elements in the parameter set that are not quantities or properties of the system. The latter also has pragmatic implications concerning, e.g., how the many parameters in a geohazard system could be





constrained in a geohazard assessment.

We went into details to show that predictions, and in turn UQ, are conditional on the model(s) chosen together with the assumptions made by analysts. Based on the identified limitations of measured data to support the assessment of the quality of predictions, we have proposed that the quality of UQ is to be judged based on crucial tasks such as the exhaustive scrutiny of the knowledge coupled to the assessment of deviations of those assumptions made in the analysis.

Key to enacting the proposed clarifications and simplifications is the full consideration of knowledge-based probabilities. Based on the proposed examination of strength of knowledge, knowledge-based probabilities can be assigned. Considering this type of probabilities will also help overcome the identified limitations of the maximum entropy principle or counterfactual analysis to quantify uncertainty in input quantities. We have exposed that the latter approaches are prone to produce unexhausted uncertainty quantification due to their reliance on measured

data, which can miss crucial events or overlook relevant input quantities.

**Appendix**

In this Appendix, the necessary details of the original analysis made by Zhao et al. (2021) are given. The following are the basic equations $E_m$ used by these authors.

$$p(y = c) \sim \frac{\sum_{x_s \in Y} \rho_{x=c, y=c}}{\sum_{c=1}^{C} \sum_{x_s \in Y} \rho_{x=c, y=c}} \tag{A-1}$$

$$\rho_{xy} = exp\left(-\pi \frac{\overline{s_x s_y}}{\theta_h} - \pi \frac{|s_x s_y|}{\theta_v}\right) \tag{A-2}$$

where $X$ is the collection of all quantities at borehole-points, which take values $x$, $Y$ is the collection of all quantities at non-borehole-points with values $y$, $\rho_{xy}$ is the value of correlation between a quantity value $x$ at a penetrated point $s_x \in S_x$ and the value $y$ at a non-penetrated point $s_y \in S_y$. $\overline{s_x s_y}$ is the horizontal distance between points $s_x$ and $s_y$, while $|s_x s_y|$ is the vertical one. $\theta_h$ and $\theta_v$ are the horizontal and vertical scale of fluctuation, respectively. Each material class considered is associated exclusively with an element in the set of integers $\{1,2,…,C\}$. $p(y=c)$ is the

probability of encountering a type of material $c$ in a point $s_y$. Such probability is initially approximated using Eq. A-1. More accurate probabilities are computed on the basis of the repeated sampling of the joint distribution $f(x,y)$, which was approximated using Eq. A-1. Eq. A-1, described in short, approximates probabilities as the ratio of the sum of correlation values calculated for a penetrated point in the set $S_x$ and the set of non-penetrated points $S_y$ for a given material $c$ to the sum of correlation values for all points and all materials.

Based on the stratigraphy data collected at borehole locations, the selection of the type of correlation function and the scales of fluctuation took place using the maximum likelihood method. The authors considered three types of correlation functions, namely squared exponential, single exponential, and second-order Markov. In this case, the likelihood function $\mathcal{L}(\theta_m|d) = f(d|\theta_m)$ represents the likelihood of observing $d$ at borehole locations, given the spatial correlation structure $\theta_m$. The squared exponential function yielded the maximum likelihood when the

horizontal scale of fluctuation and the vertical scale of fluctuation are set to 11.1 and 4.1 metres, respectively. Hence, the squared exponential function correlation whose expression is Eq. A-2 in this Appendix was selected. Eq. A-3 and A-4 correspond to the single exponential and second-order Markov functions, respectively.



$$\rho_{xy} = exp\left(-2\frac{\overline{s_x s_y}}{\theta_h} - 2\frac{|s_x s_y|}{\theta_v}\right) \quad \text{(A-3)}$$

$$\rho_{xy} = \left(1 + 4\frac{\overline{s_x s_y}}{\theta_h}\right)\left(1 + 4\frac{|s_x s_y|}{\theta_v}\right)exp\left(-4\frac{\overline{s_x s_y}}{\theta_h} - 4\frac{|s_x s_y|}{\theta_v}\right) \quad \text{(A-4)}$$

**Data availability**

The research reported in this paper did not generated data.

**Code availability**

The research reported in this paper did not generate any code.

**Authors' contributions**

Ibsen Chivata Cardenas: Conceptualization, Methodology, Writing- Original draft preparation, Investigation, Validation; Terje Aven: Investigation, Supervision, Writing- Reviewing and Editing; Roger Flage: Investigation, Supervision, Writing- Reviewing and Editing.

**Conflict of interest/Competing interests**

The authors declare that they have no conflict of interest.

**Funding**

This research is funded by ARCEx partners and the Research Council of Norway (grant number 228107).

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
