# Peer review of "Addressing challenges in uncertainty quantification. The case of geohazard assessments"

_Geoscientific Model Development, 2022_

## Author Response (AR1)

REVIEWER 1: «The paper reviews the state of the art and open challenges in uncertainty quantification for geohazard assessments. It puts a significant portion of literature into context, followed by a case-study example based on the work of Zhao et al. (2021).

The manuscript is well written, and I could not identify obvious errors. The scientific novelty is fulfilled, as the paper sheds new light on the discussion of UQ in the present context, and puts previously disconnected literature into context. The example is illustrative and well discussed.

To sum up, I can recommend the publication of the manuscript.»

AUTHORS: We appreciate the generous words. We are thankful for the recommendation for publication.

REVIEWER 2: «The authors consider the problem of quantifying uncertainty in the context of geohazard risk assessments. By studying the common challenges faced in current approaches to this problem, the authors formulate a list of seven proposed mitigations which may lead to more credible and trustworthy predictions in this area. Various results in the current literature are reinterpreted using the authors' new framework, and it is shown that relevant geohazard uncertainties can be understood as probabilities conditioned on natural priors resulting from their proposed considerations.

I have a mixed opinion of this paper. Scientifically, I believe it is of high quality, providing a useful framework in which to view uncertainty quantification along with useful insights gleamed from various works in the current literature.»

AUTHORS: We appreciate the generous words.

REVIEWER 2: «(…) Linguistically, I found it quite difficult to read even as a native speaker of English. I believe this is due to the repeated (mis)use of lengthy compound sentences and relatively complicated rhetorical devices which are easy to confuse and do not add value to the exposition. (…)»

AUTHORS: The paper was subjected to an exhaustive style correction by the main author and then by the co-authors. The authors went through the text, revising sentence by sentence. The modifications also comprised reducing the use of commas and reorganising the sentences to have a conventional order of subject-verb-object-adverb. The modifications made are highlighted in red in the revised text. When a set of letters between a couple of words is highlighted in red, it means that the original words in between were eliminated.

REVIEWER 2: «(…) I do recommend this work is eventually published. (…)»

AUTHORS: We are thankful for the recommendation for publication.

REVIEWER 2: «(…) but I strongly suggest that the authors go carefully through their exposition and aggressively simplify their sentence structure. This will have a dual benefit: not only are simple sentences clearer to the reader (therefore increasing the potential readership), (…)»

AUTHORS: We refer to a previous answer in this respect.

REVIEWER 2: «(…) they also provide less room for confusing grammatical errors which I noticed in many places, particularly in the placement of commas.  An example of this simplification is provided below, as well as more specific comments which may be useful to the authors during the revision process.  Note that my comments do not provide an exhaustive list of corrections to all of the linguistic mistakes I noticed. (…)»

AUTHORS: In the revision of the paper, we also took the opportunity to check for inadvertent grammatical errors. The modifications also comprised reducing the use of commas. The changes made have been also highlighted in red throughout the text.

REVIEWER 2: «(…) ---  Here is an example of what I mean by sentence simplification.  In lines 307-310, it is written,

``Analysts, for example, may simplify the analysis when, according to the scrutiny of $Òš$, increased knowledge about, e.g., $X\_1$ will not result in increased knowledge about another quantity, e.g., $X\_2$, and, for example, if a distribution $f(y|x\_1,x\_2)$ is to be specified, we may have then that $f(y|x\_1,x\_2)$ reduces to $f(y|x\_1)f(y|x\_2)$, according to probability theory.''

This sentence is very difficult to read and could easily be replaced with the equivalent sentence,

``For example, when increased knowledge about a quantity (say $X\_1$) will not result in increased knowledge about another quantity (say $X\_2$), analysts may simplify the analysis according to the scrutiny of $K$, meaning that a distribution $f(y|x\_1,x\_2)$ to be specified may reduce to $f(y|x\_1)f(y|x\_2)$ according to probability theory.''

Note that this carries the same meaning while reducing the amount of unnecessary commas by 9! (…)»

AUTHORS: The suggested changes are highlighted within text lines 295-297 in the revised paper. We have also simplified other sentences throughout the manuscript.

REVIEWER 2: «(…) --- Stylistically, I think it would be good to put commas at the end of the broken-out equations when they appear in the middle of a sentence, and periods when they appear at the end.  This way the lines of equations feel connected to the surrounding exposition(…)»

AUTHORS: The proposed changes are highlighted in red throughout the text. The text lines 72-75 provides an example.

REVIEWER 2: «(…) --- The text in Tables 5 and 6 should not be indented.(…)»

AUTHORS: The required changes were made in Table 5 in parts a and b.

REVIEWER 2: «(…) --- Please check that the notation in section 4 is consistent with that in section 2, specifically regarding bolded/unbolded variables.(…)»

AUTHORS: Checks were undertaken for consistency in the use of bolded and unbolded quantities. The detected inconsistencies were corrected. The corrections are highlighted in red in the text lines 366, 411, 447, and Table 4

REVIEWER 2: «(…) --- Data availability statement: ``generated'' should be replaced with ``generate any''. (…)»

AUTHORS: The requested change has been made in text line 519.

REVIEWER 2: «(…) --- Line 240: The phrase ``on the basis on the scrutiny'' should be replaced with ``based on the scrutiny''. (…)»

AUTHORS: The requested change has been made in text line 231.

REVIEWER 2: «(…) --- Line 283:  I recommend ``the maximum entropy principle'' instead of ``the principle''. (…)»

AUTHORS: The requested change has been made in text line 148.

REVIEWER 2: «(…) --- Line 306:  I recommend removing the comma after $X_2$ (…)»

AUTHORS: The requested change has been made in text line 292.

REVIEWER 2: «(…)--- Line 323:  Should $\theta_\varepsilon$ symbol be in bold font? (…)»

AUTHORS: We agree with the reviewer. We modified the text accordingly. The requested change has been made in text line 310.

REVIEWER 2: «(…) --- Line 356:  The phrase `We use an analysis and information by Zhao et al. (2021), which refer to'' should be replaced with ``We use analysis and information from Zhao et al. (2021), which refers to''.»

AUTHORS: The requested change has been made in text line 342.

---

## Author Response (AR2)

Dear Editor
Dan Lu
Handling topical editor
Geoscientific Model Development

Good afternoon. Again many thanks for the consideration of our proposed paper. We have already addressed the suggested corrections by the reviewer. These are now highlighted in yellow in the below text. All the changes made are located in the same text lines identified by the reviewer. We accepted all the suggestions. Below is a transcription of a copy of the reviewer report.

"(…) (visible to the public if the article is accepted and published)

This paper offers an interesting viewpoint on the credibility of uncertainty quantification and its role in geohazard risk assessment. I found the authors' seven considerations to be thought provoking, and their survey of the literature to be useful and reasonably comprehensive. I recommend their work for publication provided the following minor grammatical corrections are addressed.

Abstract: change "recently developed sophisticated" to "recently developed, sophisticated".
Line 117: remove the indentation.
Line 198: change "repeated tests make" to "repeated tests, make".
Line 199: change "can be suggested" to "suggests".
Line 208: change "and models M and" to "and models M, and".
Line 214: change "…, and when knowing x and \theta, …" to "… given knowledge of x and \theta, …".
Line 246: change "assessed, mainly" to "assessed mainly"
Line 275: remove "among others,".
Line 292: remove the comma after X_2.
Line 303: change "take" to "takes".
Line 309-312: This sentence should be turned around. I recommend the following change. "For example, analysts may consider that parameters \Theta_\varepsilon are not part of the system as such, since they are linked to: the output-prediction error \varepsilon, some model parameters in \Theta_m, the vector associated with observation/measurement errors \Theta_o, and the overall attached hyperparameters linked to probability distributions (including priors and likelihood functions).
Line 361: change "values y and x and" to "values y and x, and".
Table 4: change "X only take values" to "X only takes values".
Line 387: change "but still useful" to "but is still useful".
Tables 5 and 6: missing periods at the end of many sentences.
Line 423: change "The observation" to "This observation".
Lines 440-441: I find this sentence very hard to understand as written. It would be more clear to say something like the following. "Accordingly, further examination of the existing knowledge K justifies some assessment of the impact of assuming a local rather than global scale of fluctuation."
Lines 445-449: All semicolons should be replaced with commas.
Line 450: change "somewhat an agreement on" to "some agreement on".
Line 454: remove the word "other". (…)

Best regards,

Ibsen Chivata Cardenas
Corresponding author

[revised manuscript text omitted]